# Sorghum Densification with Changes in Plant Spacing Arrangement: Productivity and Qualitative Characteristics of Silage Material

**Dayenne M. Herrera** [1], **Wender M. Peixoto** [1,*], **Joadil G. de Abreu** [1], **Rafael H. P. dos Reis** [2], **Carlos E. A. Cabral** [3], **Livia V. de Barros** [4], **Vanderley A. C. Klein** [2] and **Edmilson F. dos Passos** [2]

[1] Graduate Program in Tropical Agriculture, Faculty of Agronomy and Animal Sciences, Federal University of Mato Grosso, Cuiabá 78060-900, MT, Brazil; dayenne.herrera@gmail.com (D.M.H.); joadil.abreu@ufmt.br (J.G.d.A.)

[2] Federal Institute of Education, Science and Technology of Rondônia, Colorado do Oeste 76993-000, RO, Brazil; rafael.reis@ifro.edu.br (R.H.P.d.R.); vanderley.klein@ifro.edu.br (V.A.C.K.); edmilson.fabiciakc@gmail.com (E.F.d.P.)

[3] Institute of Agricultural and Technological Sciences, Federal University of Rondonópolis, Rondonópolis 78736-900, MT, Brazil; carlos.eduardocabral@hotmail.com

[4] Institute of Agricultural Sciences, Federal University of Minas Gerais, Montes Claros 39404-547, MG, Brazil; liviavieiradebarros@gmail.com

* Correspondence: wender.mpeixoto@gmail.com

**Abstract:** The aim of this study was to evaluate the agronomic performance of sorghum grown in different combinations of row spacing and plant density, as well as possible interferences on silage quality. No other study dedicated to identifying the interference of plant spatial arrangement on the cultivation of silage material has been developed in the productive context of the Amazon Biome, making it necessary to understand the behavior of the studied factors. The treatments were arranged in a split-plot scheme: the plots corresponded to three row spacings (0.45 m, 0.60 m, and 0.75 m) and subplots at four densities (105,000, 120,000, 135,000, and 150,000 plants ha$^{-1}$). The agronomic and productivity characteristics of sorghum and the fermentative and bromatological characteristics of forage and silage were evaluated. The sorghum plants showed an increase in plant height and green and dry mass yield when using higher densities ($p < 0.05$). For the culm diameter variable, an isolated effect of the factors was observed, with reduced diameter when grown closer to inter-row spacing or using higher plant densities. No effect of the factors was found ($p > 0.05$) for morphological plant components. In silage, wider spacing promoted higher dry matter content. Regarding crude protein in the silage, higher percentages were obtained at closer spacing and higher plant density. The sorghum growing in dense conditions is indicated, given the positive performance in productivity and bromatological composition.

**Keywords:** chemical composition; plant population; row spacing; *Sorghum bicolor*; yield

## 1. Introduction

The Brazilian cattle herd's diet is based on pasture formed with forage grasses. Despite the dry mass production potential, these grasses present seasonal production [1–3].

During the dry season, to minimize the lack of fulfilling the animals' nutritional needs due to the low availability and quality of roughage, alternatives are needed for forage production and conservation. From this perspective, the use of silage is an excellent option, providing dry matter, energy, and conserved nutrients [4–7].

Although different forage plants can be used for silage production, the sorghum (*Sorghum bicolor* (L.) Moench.) crop is of particular interest since it has similar nutritional value to corn (*Zea mays* L.) with good yield potential and a greater tolerance to water deficit, amplitude of the sowing season, and regrowth capacity [8,9].

Sorghum used for silage is selected according to its green mass yield, and its quality as a roughage feed is influenced by the proportion of grains. It can be seen, however, that

these productive components are directly affected by the sowing conditions to which the crop is subjected [10,11] such as the plant arrangement defined by plant spacing and density.

Overall, the combination of inter-row spacing and sowing density positively influences grain yield, mainly through the number of available panicles. The density determines the distribution of plants per area and, consequently, is conditioned to different sunlight interceptions [10], a fact that can promote changes in plant composition, as a result of adjustments in vegetative growth. It is suggested that population arrangements, which have an influence on morphological characteristics, can either favor or hinder performance regarding the desired qualitative aspects for materials dedicated to silage. Therefore, it is necessary to identify the spatial dispositions in which the plants intended for silage can be submitted during their growing.

Thus, the aim of this study was to verify how the plant arrangement, with adjustments to inter-row spacing and sowing densities, influences the agronomic performance and characteristics of sorghum silage.

This study holds significant relevance regarding the cultivation method for forage crops used as silage. Within the productive context embedded in the Amazon Biome, no other work in this field has been undertaken. For this reason, the originality of this study highlights the necessity and impact of the presented information.

## 2. Materials and Methods

### 2.1. Experimental Site

The experiment was carried out in the vegetable production area of the Federal Institute of Education, Science and Technology of Rondônia (IFRO), located in Colorado do Oeste, Rondônia, Brazil (13°07′39″ S; 60°29′68″ W; 460 m altitude). The soil is classified as Eutrophic Red Argisol, with clay texture and flat-wavy slope gradient. A soil analysis performed on the 0 to 20 cm layer revealed the following chemical characteristics: pH (CaCl$_2$) = 5.20; organic matter = 20.0 g dm$^{-3}$; $p$ = 5.0 mg dm$^{-3}$; K = 2.1 mmol$_c$ dm$^{-3}$; Ca = 48.8 mmol$_c$ dm$^{-3}$; Mg = 8.3 mmol$_c$ dm$^{-3}$; H + Al = 25.0 mmol$_c$ dm$^{-3}$; cation exchange capacity = 84.0 mmol$_c$ dm$^{-3}$; and base saturation = 59.0%.

The local climate is classified as tropical monsoon (Am) according to the Koppen–Geiger classification, with two well-defined seasons. The total rainfall during the experimental season was 1432.6 mm, with an average temperature of 23.7 °C (Figure 1).

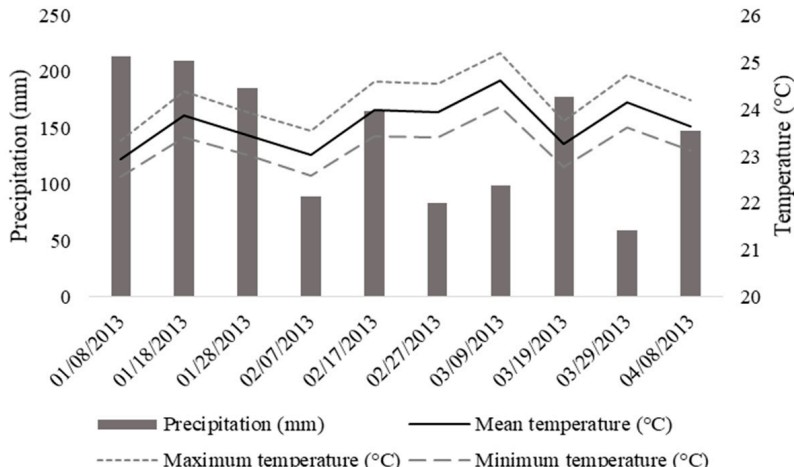

**Figure 1.** Precipitation and temperature every 10 days during the experiment [12].

### 2.2. Experimental Design and Management

The experimental design used was in randomized blocks, with four repetitions. The treatments were arranged in a split-plot scheme: the plot corresponded to three inter-row spacings (0.45 m, 0.60 m, and 0.75 m), and the subplots were at four sowing densities

(105,000, 120,000, 135,000, and 150,000 plants ha$^{-1}$). The experimental unit was composed of five rows 5.0 m long, with the three central rows being the usable area, disregarding 1.0 m from the ends. Usable areas of 4.05 m$^2$, 5.4 m$^2$, and 6.76 m$^2$ were harvested in plots spaced at 0.45 m$^2$, 0.60 m$^2$, and 0.75 m$^2$, respectively.

The sorghum hybrid BRS 655 was used, which has a tall stature and is commonly used for forage and silage production. Sowing was performed manually, with fertilization at doses of 20 kg ha$^{-1}$ of N, 80 kg ha$^{-1}$ of P$_2$O$_5$, and 60 kg ha$^{-1}$ of K$_2$O, using urea, triple superphosphate, and potassium chloride, respectively. Five days after sowing, uniformity in the emergence of the seedlings was observed, and 17 days after emergence, nitrogen fertilization was applied in cover at a dose of 100 kg ha$^{-1}$.

The monitoring of pests, diseases, and weeds was carried out and, when the acceptable level was reached, management practices were adopted. To control the fall armyworm (*Spodoptera frugiperda*) and leafhopper (*Empoasca kraemeri*), it was necessary to apply an insecticide based on Thiamethoxam and Lambda-cyhalothrin at a dose of 0.15 L ha$^{-1}$. Weed control was carried out with manual weeding.

### 2.3. Measurements

An agronomic characteristics evaluation was carried out before plant harvesting for silage. Ten plants were used from the useful area, and plant height (HT, m), culm diameter at 20 cm from the ground (DIA, cm), and panicle length (PANL, cm) were determined. Subsequently, the plants were harvested at 20 cm from the ground and, to determine the green leaf (GLP, %), leaf litter (LLP, %), culm (CULP, %), and panicle (PANP, %) proportions, the components were fractioned and weighed separately, relating them to the plant dry mass.

The green mass yield (GMY, t ha$^{-1}$) was calculated from the total mass of plants harvested from each plot in a known area. For dry mass yield (DMY, t ha$^{-1}$) estimation, a correction was made on the basis of forage dry matter. Harvesting was carried out according to the silage point recommendation for sorghum, i.e., when the grain was in the early dough stage around 30–35% dry matter. Harvesting was performed manually at 20 cm above the soil surface. The forage was chopped using a stationary chopper, with particle size between 2.0 and 3.0 cm.

### 2.4. Ensiling and Assessment of Silage Samples

The experimental units in the field were experimental silos (glass jars) with a volume of 2.5 L, equipped with a lid for sealing and adapted with a "siphon" type valve to allow the exit and prevent the entry of gases inside the silo. The experimental silos were filled with enough forage for their complete filling after compaction, obtaining an average density of 640 kg m$^{-3}$ of green mass. After filling, the silos were closed and sealed, and stored for 30 days in a ventilated place away from direct light.

At the time of ensiling, forage samples were collected. A subsample was used to determine the buffer capacity (BC, eq. mg HCl 100 g$^{-1}$ of DM) using the methodology proposed by Mizubuti et al. [13]. Upon opening of the experimental silos, silage samples were also collected.

The forage and silage samples were dried using the freeze-drying method, from sublimation to −58 °C for 72 h, and then milled in a Willey mill on 1.0 mm mesh screens. The dry matter (DM, %) content was quantified using the AOAC [14] method.

A pH analysis was performed following the method proposed by Silva and Queiroz [15] using a bench-top pH meter. The ammoniacal nitrogen (NH$_3$-N, %) content was quantified from the aqueous extract at a ratio of 1:2, distilled with 2N KOH in a Micro-Kjeldahl apparatus [16].

The nitrogen content was determined using the Kjeldahl method and multiplied by the coefficient of 6.25 to obtain the crude protein concentrations [14]. The neutral detergent fiber was analyzed in a detergent solution [17] with autoclave extraction [18].

*2.5. Statistical Analysis*

The Shapiro–Wilk test was used to assess the normality of the residual data. All variables followed a normal distribution except LLP, with an arc-sine transformation being performed. The data were subjected to an analysis of variance using the Experimental Designs package of R Software, version 3.5.3 [19], with spacing, population, and their interaction as the main effects. Statistical significance was set to $p < 0.05$, and the mean differences were analyzed using Tukey's multiple comparison test for the qualitative factor and regression analysis for the quantitative factor.

## 3. Results

The inter-row spacing did not affect the PANL or the GLP, LLP, CULP, or PANP fractions ($p > 0.05$; Table 1). Similarly, there was no effect of plant population on these variables. The hybrid used seems to adapt, according to its components, to the different sowing space arrangements.

**Table 1.** Summary of the analysis of variance for agronomic characteristics of sorghum (*Sorghum bicolor* (L.) Moench.) as a function of row spacing and plant population in Colorado do Oeste, RO, Brazil.

| Parameter | Inter-Row Spacing (m) | | | Plant Density (Plants ha$^{-1}$) | | | | *p*-Value | | | s.e. |
|---|---|---|---|---|---|---|---|---|---|---|---|
| | 0.45 | 0.60 | 0.75 | 105,000 | 120,000 | 135,000 | 150,000 | S | D | I | |
| **Agronomic Characteristics** | | | | | | | | | | | |
| PANL (cm) | 23.17 | 23.59 | 24.91 | 26.29 | 22.74 | 25.05 | 21.50 | 0.45 | 0.10 | 0.15 | 5.20 |
| GLP (%) | 12.15 | 11.90 | 11.13 | 11.88 | 11.20 | 12.52 | 11.32 | 0.49 | 0.47 | 0.19 | 2.31 |
| LLP (%) | 2.85 | 2.80 | 2.63 | 3.00 | 2.38 | 3.03 | 2.61 | 0.92 | 0.28 | 0.24 | 1.18 |
| CULP (%) | 43.21 | 40.69 | 40.47 | 41.31 | 42.16 | 41.74 | 40.62 | 0.37 | 0.94 | 0.29 | 6.90 |
| PANP (%) | 41.77 | 44.59 | 45.76 | 43.79 | 44.24 | 42.69 | 45.43 | 0.07 | 0.83 | 0.37 | 7.20 |

PANL: panicle length (cm); GLP: green leaf proportion (%); LLP: leaf litter proportion (%); CULP: culm proportion (%); PANP: panicle proportion (%); S: subjective interpretation; D: descriptive interpretation; I: inferential interpretation; s.e.: standard error.

There was no interaction effect between plant spacing and density on HT, DIA, GMY, or DMY ($p > 0.05$; Table 2). A linear effect of plant density on these variables was observed. Using higher densities, HT, GMY, and DMY increased, while DIA decreased.

The inter-row spacing had an effect ($p < 0.05$) only on DIA. The use of narrower spacing promoted higher DIA (1.90 cm), while the 0.60 and 0.75 m spacing promoted smaller diameters.

For forage, no effect of inter-row spacing or plant density was observed on pH, BC, DM, CP, or NDF (Table 3). There was also no effect of the factors on the variables pH, $NH_3$-N, or NDF of the silage.

There was an isolated effect of inter-row spacing ($p < 0.05$) on the DM percentage of silage (Table 4). The DM decreased considerably when the wider spacing (0.75 m) was used, while the narrower spacing (0.45 m) promoted a higher dry matter percentage.

A significant interaction effect between inter-row spacing and plant density was observed on the CP of silage ($p < 0.05$). The CP decreased quadratically ($p < 0.05$) with increasing plant density associated with the 0.75 m inter-row spacing, but there was no effect of the other spacings. The inter-row spacing, in turn, had an effect on CP when associated with a population of 150,000 plants ha$^{-1}$, with a significant reduction in the crude protein percentage when using wider spacing.

**Table 2.** Total height, culm diameter, and yield of sorghum (*Sorghum bicolor* (L.) Moench.) grown at different spacing and sowing densities in Colorado do Oeste, RO, Brazil.

| Inter-Row Spacing (m) | Plant Density (Plants ha$^{-1}$) | | | | Effect | Mean | s.e. |
|---|---|---|---|---|---|---|---|
| | **105,000** | **120,000** | **135,000** | **150,000** | | | |
| | HT (m) | | | | | | |
| 0.45 | 2.21 | 2.24 | 2.27 | 2.31 | | 2.26 | 0.05 |
| 0.60 | 2.20 | 2.22 | 2.26 | 2.27 | | 2.24 | 0.06 |
| 0.75 | 2.20 | 2.25 | 2.26 | 2.26 | | 2.24 | 0.05 |
| Mean | 2.21 | 2.24 | 2.26 | 2.28 | L (<0.05) | | |
| s.e. | 0.08 | 0.05 | 0.03 | 0.03 | | | |
| | DIA (cm) | | | | | | |
| 0.45 | 2.03 | 1.98 | 1.83 | 1.78 | | 1.90 a | 0.13 |
| 0.60 | 1.78 | 1.60 | 1.63 | 1.59 | | 1.73 b | 0.09 |
| 0.75 | 1.78 | 1.76 | 1.67 | 1.66 | | 1.65 b | 0.08 |
| Mean | 1.86 | 1.78 | 1.71 | 1.68 | L (<0.05) | | |
| s.e. | 0.14 | 0.17 | 0.10 | 0.10 | | | |
| | GMY (kg ha$^{-1}$) | | | | | | |
| 0.45 | 43,674 | 45,537 | 49,033 | 50,990 | | 47,308 | 4600 |
| 0.60 | 41,450 | 46,005 | 45,731 | 48,630 | | 45,454 | 3838 |
| 0.75 | 42,875 | 43,852 | 46,213 | 51,782 | | 46,181 | 4077 |
| Mean | 42,666 | 45,132 | 46,992 | 50,467 | L (<0.05) | | |
| s.e. | 3856 | 2123 | 3429 | 2768 | | | |
| | DMY (kg ha$^{-1}$) | | | | | | |
| 0.45 | 12,860 | 13,600 | 15,274 | 15,093 | | 14,207 | 1505 |
| 0.60 | 12,170 | 13,349 | 13,987 | 16,203 | | 13,927 | 1962 |
| 0.75 | 12,342 | 14,064 | 14,342 | 16,672 | | 14,355 | 1916 |
| Mean | 12,457 | 13,671 | 14,534 | 15,989 | L (<0.05) | | |
| s.e. | 1050 | 917.47 | 1346 | 1581 | | | |

HT: plant height (m); DIA: culm diameter (cm); GMY: green mass yield (kg ha$^{-1}$); DMY: dry mass yield (kg ha$^{-1}$). The inter-row spacing means within culm diameter (DIA) are denoted by different letters in the column, indicating variations between them ($p < 0.05$); L: observed significance level for linear effects; s.e.: standard error.

**Table 3.** Summary of the analysis of variance for forage and silage characteristics of sorghum (*Sorghum bicolor* (L.) Moench.) as a function of inter-row spacing and plant population in Colorado do Oeste, RO, Brazil.

| Parameter | Inter-Row Spacing (m) | | | Plant Density (Plants ha$^{-1}$) | | | | *p*-Value | | | s.e. |
|---|---|---|---|---|---|---|---|---|---|---|---|
| | **0.45** | **0.60** | **0.75** | **105,000** | **120,000** | **135,000** | **150,000** | **S** | **D** | **I** | |
| Forage characteristics | | | | | | | | | | | |
| pH | 5.93 | 5.93 | 5.95 | 5.94 | 5.93 | 5.94 | 5.94 | 0.86 | 0.93 | 0.21 | 0.07 |
| BC (eq. mg) | 18.00 | 18.31 | 17.55 | 18.81 | 17.05 | 18.40 | 17.57 | 0.49 | 0.28 | 0.61 | 2.30 |
| DM (%) | 30.08 | 30.56 | 31.07 | 29.31 | 30.34 | 30.91 | 31.72 | 0.58 | 0.08 | 0.22 | 2.56 |
| CP (%) | 8.17 | 8.56 | 8.02 | 8.18 | 8.31 | 8.13 | 8.39 | 0.17 | 0.78 | 0.13 | 0.71 |
| NDF (%) | 53.08 | 50.81 | 51.29 | 52.67 | 50.70 | 51.91 | 51.62 | 0.21 | 0.75 | 0.43 | 4.37 |
| Silage characteristics | | | | | | | | | | | |
| pH | 3.79 | 3.80 | 3.90 | 3.87 | 3.83 | 3.82 | 3.80 | 0.06 | 0.58 | 0.09 | 0.15 |
| NH$_3$-N (% TN) | 3.37 | 3.48 | 3.98 | 3.77 | 3.55 | 3.49 | 3.63 | 0.27 | 0.93 | 0.10 | 1.17 |
| NDF (%) | 58.57 | 56.01 | 54.04 | 56.59 | 57.24 | 56.60 | 54.41 | 0.09 | 0.29 | 0.50 | 4.28 |

pH: potential of hydrogen; BC: buffer capacity (eq. mg. HCl 100 g$^{-1}$ of DM); DM: dry matter (%); CP: crude protein (% of DM); NDF: neutral detergent fiber (% of DM); NH$_3$-N: ammoniacal nitrogen (% of TN); S: subjective interpretation; D: descriptive interpretation; I: inferential interpretation; s.e.: standard error.

**Table 4.** Dry matter and crude protein of silage sorghum (*Sorghum bicolor* (L.) Moench.) grown at different spacing and sowing densities in Colorado do Oeste, RO, Brazil.

| Inter-Row Spacing (m) | Plant Density (Plants ha$^{-1}$) | | | | Effect | Mean | s.e. |
|---|---|---|---|---|---|---|---|
| | 105,000 | 120,000 | 135,000 | 150,000 | | | |
| | | | DM (%) | | | | |
| 0.45 | 27.99 | 27.76 | 26.47 | 28.09 | | 29.42 a | 2.82 |
| 0.60 | 28.42 | 28.14 | 29.28 | 29.43 | | 28.82 ab | 2.24 |
| 0.75 | 28.83 | 29.46 | 30.04 | 29.34 | | 27.58 b | 1.91 |
| Mean | 28.41 | 28.45 | 28.60 | 28.96 | ns | | |
| s.e. | 1.94 | 3.03 | 2.40 | 2.52 | | | |
| | | | CP (% of DM) | | | | |
| 0.45 | 7.81 a | 7.89 a | 8.56 a | 8.04 a | ns | 8.07 | 0.33 |
| 0.60 | 8.59 a | 8.22 a | 7.52 a | 7.55 ab | ns | 7.97 | 0.52 |
| 0.75 | 7.82 a | 8.67 a | 7.58 a | 6.75 b | Q (<0.05) | 7.70 | 0.78 |
| Mean | 8.07 | 8.26 | 7.89 | 7.45 | | | |
| s.e. | 0.44 | 0.39 | 0.58 | 0.64 | | | |

Means followed by the same letter in the column do not differ statistically ($p < 0.05$); ns: not significant. Q: observed significance level for quadratic effects. DM: dry matter (%); CP: crude protein (% of DM); s.e.: standard error.

## 4. Discussion

This study provided interesting information about the effects of different sowing space arrangements on the growth of silage sorghum. The results suggest that the hybrid used seems to adapt to the use of different inter-row spacing and plant densities since the plant densities combinations did not alter the morphological fraction of the plants (Table 1). The same behavior was confirmed by other researchers [20,21], evidencing the good structural maintenance potential of plants under these growing conditions.

The sorghum plant must present a balance among the culm, leaf, and panicle fractions, considering not only the productive aspects but also the nutritional value. The mean value of the PANP fraction (44.04%) stood out over the other plant components, conferring good participation [22] within the context of producing high-quality silage. The same authors suggest that the greater the grain proportion present, the better the fermentative and nutritional quality of the silage.

### 4.1. Plant Height, Culm Diameter, and Mass Yield

The plant height increased under higher plant densities, which was due to the combined effect of intra-specific competition for light, apical dominance stimulation, and resource reallocation [23–25]. This behavior was observed by Carmo et al. [10] when evaluating the agronomic performance of sorghum sown in double-row spacing in the Cerrado, indicating that with increasing sorghum sowing density, there was an increase in HT, just as it was observed in this study.

Regarding the culm diameter, there was a tendency for the thickness to decrease when cropped with larger populations. Therefore, culm diameter and plant height correlate inversely proportionally given the use of higher densities. The observations suggest that plant density in the sowing row may result in the sorghum plants being more exposed to lodging, requiring more attention to choose adequate genotypes.

The DIA, on the other hand, had a tendency toward greater thickness when narrower inter-row spacing was used. Smaller spacings optimize radiation interception because of the better spatial distribution of the plants in the leaf area [26], a fact that may justify the development of culm under these growing conditions.

The green and dry mass yields were closely related to plant density. The linear response effect suggested differences of 18.28 and 28.35% for green and dry mass, respectively, using the higher plant density. The increase in GMY and DMY, given the densification, defines the hybrid adaptability to higher sowing density. Under the same conditions, greater HT was observed, which may have contributed to the increase in mass in addition

to the greater number of plants per area under denser conditions. Avelino et al. [21] found no effect of density on productivity when evaluating the yield performance and agronomic characteristics of sorghum hybrids AG-2005 and AGX-213, grown at 0.50 m, 0.75 m, and 1.00 m inter-row spacing sowing.

### 4.2. Forage Characteristics

The mean value of BC is suitable for the forage characteristics indicated for silage [27], making it possible to reduce the forage pH from the fermentative processes inside the silo. According to McCullough [28], the ideal fermentation in a silo is expected when the ensiled forage has a dry matter content between 28 and 34%, suggesting, therefore, adequacy of the mean DM content obtained in this study.

The mean CP content obtained from the forage is in accordance with the nutritional requirements of cattle, given the supply of sufficient nitrogen for adequate rumen development. The mean values found for NDF contents are close to those reported by Albuquerque et al. [29] and Avelino et al. [30], who also observed no effect of spacing and density in forage sorghum cropping.

### 4.3. Silage Characteristics

The pH values lower than 3.90 and $NH_3$-N lower than 3.98% of total N indicate that good fermentation occurred inside the silo, with adequate pH reduction in the ensiled forage and low action of undesirable microorganisms, such as bacteria of the genus *Clostridium*, which degrade protein [27]. The mean values observed are similar to those reported by Tolentino et al. [31] for the same forage hybrid, suggesting that the fermentative pattern is not influenced by the combinations of the spatial arrangement of plants.

The fibrous fraction, represented by NDF, maintained participation similar to other works with forage sorghum [31]. Avelino et al. [30], on the contrary, obtained a reduced NDF content as a function of densification and associated this effect with the dilution of the culm in the total plant composition.

The reduced inter-row spacing promoted an increase in DM contents. This result corroborated the observations of Avelino et al. [30] who confirmed a plant's ability to better synthesize photo-assimilates as a function of an appropriate arrangement for light interception.

The combination of larger spacings and higher densities promoted a quadratic behavior with a slight reduction in CP. Under the same growing conditions, culm elongation has a low CP content and therefore may have acted as a dilution effect.

In view of the research already carried out, it is known that inter-row spacing is still very varied, notwithstanding the current adaptation of harvesters to spacing of up to 0.45 m allows farmers to use reduced spacing.

From this perspective, it is worth considering the need to verify the economic feasibility of implementing and cropping different plant spacings and densities, as well as their respective return on productive yield, which varies according to the production system used by a rural property.

### 4.4. Limitations of the Study

With regard to the fundamental principles of experimentation, such as replication, randomization, and local control, special attention was given to their implementation in this study. On the other hand, replication of the trials across different agricultural years could not be applied during the course of this study. Conducting the experiment in only one growing season may compromise the reliability of the results concerning potential behaviors that were not observed within the monitored soil and climatic conditions.

## 5. Conclusions

The sorghum hybrid BRS 655 adapted in height and diameter with densification in spacing and plant density under the experimental conditions of this study, without changing the plant structural composition of the plant.

The forage presents adequate composition and fermentative profile, suggesting viability in obtaining silage even with high-density cropping.

A densified sorghum plant arrangement favors dry matter and crude protein characteristics of silage.

When choosing the sowing arrangement, in addition to the implementation cost, it is necessary to consider the advantages and management facilities that it provides for adequacy in decision-making.

**Author Contributions:** Conceptualization, D.M.H. and R.H.P.d.R.; methodology, D.M.H., V.A.C.K. and E.F.d.P.; software, D.M.H.; validation, J.G.d.A., C.E.A.C. and L.V.d.B.; formal analysis, D.M.H.; investigation, D.M.H., W.M.P. and E.F.d.P.; resources, D.M.H., R.H.P.d.R. and V.A.C.K.; data curation, D.M.H. and W.M.P.; writing—original draft preparation, D.M.H. and W.M.P.; writing—review and editing, D.M.H. and W.M.P.; visualization, D.M.H., W.M.P., J.G.d.A., R.H.P.d.R., C.E.A.C., L.V.d.B., V.A.C.K. and E.F.d.P.; supervision, R.H.P.d.R.; project administration, R.H.P.d.R. All authors have read and agreed to the published version of the manuscript.

**Funding:** This research was financially supported by the Federal Institute of Education, Science and Technology of Rondônia (IFRO), through the grant provided under the document number 23243.016184/2023-77, as recorded in the electronic information system of IFRO

**Institutional Review Board Statement:** Not applicable.

**Informed Consent Statement:** Not applicable.

**Data Availability Statement:** The data used in this study are available from the corresponding authors on request.

**Conflicts of Interest:** The authors declare no conflicts of interest.

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
