# Peer review of "Sorghum Densification with Changes in Plant Spacing Arrangement: Productivity and Qualitative Characteristics of Silage Material"

_agronomy, doi:10.3390/agronomy14020358_

Round 1

Reviewer 1 Report

This paper evaluated the effects of planting density and row spacing on the agronomic performance and quality of silage sorghum. The work has certain significance, but the flaws in experimantation were also found. My major concerns were as follows:

1. Only one season data was included in this work, in most cases the field experiment must be repeated in at least two seasons or locations.

2. The novelty of the present work was not significant, as the effects of planting density and row spacing on the growth of cereal crops have been intensively studied.

3. The data in the tables were presented in a confusing way, which greatly reduce the readability of this work.

Therefore, I suggest to reject the manucript in its current form, but a resubmission with season/location replication data incorporated might be acceptable.

Reviewer 2 Report

Slight corrections of the English language throughout the paper are advised; The authors should better define the title; correct the abstract in order to  describe  the essence of the research better; to add a few more keywords  that  define the research more closely, to include more results in the conclusion part; to cite one to two references previously published in the "Agronomy" and include them in the references.

Moderate editing of the English language is required.

Reviewer 3 Report

General aspects

The article's theme is relevant from the point of view of applying the results generated to a specific region/production system. It sought to evaluate the performance of sorghum used for silage with variation in plant arrangement (row spacing and plant density). The work contributes with information for specific use in the intended cultivation conditions.

In the introduction, it is suggested to make clear the hypotheses and economic aspects associated with the change of plant arrangement.

Some methodological issues need to be complemented. For example, how was the variation in density executed? Was it estimated through the amount of seeds? Was sowing done and then thinning to adjust the densities? Were densities confirmed by counting plants after emergence?

Cultivation conditions, especially meteorological ones that occurred during the trial can be addressed in the results and discussions. It is important to assess whether the responses, especially in relation to density, would be the same under limiting water conditions. Despite the great tolerance of sorghum to water deficit, what is the expected response and risk of water deficit occurrence at the sowing date and production system used?

Some adjustments are suggested to qualify the paper: 

- Line 13: I suggest including a brief introduction on the topic.

- Line 24: reinforce that the results are applicable to sorghum used for fodder or silage production

- Line 50: include the hypotheses

- Line 97: specify the area harvested

- Line 130-131: a factor was cited as qualitative. Which factor? I understand that row spacing and plant density are both quantitative.

- Line 138: include the meaning of “S”, “D”, “I” and “s.e.”. In the other tables as well.

- Line 198: mentions lodging, but does not present data or results on the occurrence of lodging in the experiment. I suggest including and discussing it.

- Line 257: mentions the importance of considering the implementation costs when choosing the arrangement. This is a generic statement with no connection to the purpose of the work. It is important to include data, discuss and conclude with more solid bases.

Round 2

Reviewer 1 Report

No comments.